# AFP Response to Locoregional Therapy Can Stratify the Risk of Tumor Recurrence in HCC Patients after Living Donor Liver Transplantation

**DOI:** 10.3390/cancers15051551

**Published:** 2023-03-01

**Authors:** I-Hsuan Chen, Chien-Chin Hsu, Chee-Chien Yong, Yu-Fan Cheng, Chih-Chi Wang, Chih-Che Lin, Chao-Long Chen

**Affiliations:** 1Liver Transplantation Center, Department of Surgery, Kaohsiung Chang Gung Memorial Hospital, Chang Gung University College of Medicine, Kaohsiung 83301, Taiwan; 2Department of Nuclear Medicine, Kaohsiung Chang Gung Memorial Hospital, Chang Gung University College of Medicine, Kaohsiung 83301, Taiwan; 3Department of Diagnostic Radiology, Kaohsiung Chang Gung Memorial Hospital, Chang Gung University College of Medicine, Kaohsiung 83301, Taiwan

**Keywords:** alpha-fetoprotein, hepatocellular carcinoma, liver transplantation, locoregional therapy

## Abstract

**Simple Summary:**

We evaluated the effect of AFP response to locoregional therapy (LRT) on the outcomes of hepatocellular carcinoma patients after living donor liver transplantation. The enrolled patients were divided into four groups according to LRT and AFP response to LRT. The nonresponse group had the highest 5-year cumulative recurrence rates whereas the complete-response group (patients with abnormal AFP before LRT and with normal AFP after LRT) had the lowest 5-year cumulative recurrence rates among the four groups. The 5-year cumulative recurrence rates of the partial-response group (AFP response was not back to the normal level) was comparable to the control group. AFP response to LRT can be used to stratify the risk of HCC recurrence after LDLT and also to clarify the efficacy of LRT. A better prognosis can be expected if a partial AFP response of over 15% is achieved.

**Abstract:**

(1) Background: Alpha-fetoprotein (AFP) has been incorporated into the selection criteria of liver transplantation and been used to predict the outcome of hepatocellular carcinoma (HCC) recurrence. Locoregional therapy (LRT) is recommended for bridging or downstaging in HCC patients listed for liver transplantation. The aim of this study was to evaluate the effect of the AFP response to LRT on the outcomes of hepatocellular carcinoma patients after living donor liver transplantation (LDLT). (2) Methods: This retrospective study included 370 HCC LDLT recipients with pretransplant LRT from 2000 to 2016. The patients were divided into four groups according to AFP response to LRT. (3) Results: The nonresponse group had the worst 5-year cumulative recurrence rates whereas the complete-response group (patients with abnormal AFP before LRT and with normal AFP after LRT) had the best 5-year cumulative recurrence rate among the four groups. The 5-year cumulative recurrence rate of the partial-response group (AFP response was over 15% lower) was comparable to the control group. (4) Conclusions: AFP response to LRT can be used to stratify the risk of HCC recurrence after LDLT. If a partial AFP response of over 15% declineis achieved, a comparable result to the control can be expected.

## 1. Introduction

Hepatocellular carcinoma (HCC) is the sixth most commonly diagnosed cancer worldwide and the third most common cause of cancer-related deaths [1]. Liver transplantation (LT) is the best radical treatment option because it can remove the cancer as well as the entire pre-cancerous cirrhotic liver [2]. The Milan and University of California San Francisco (UCSF) transplantation selection criteria are based on morphological variables (tumor size and number), but there is increasing evidence that the biological marker alpha-fetoprotein (AFP) is a powerful predictor of tumor recurrence [3]. Recently, some centers have expanded the selection criteria or offered a downstaging protocol for advanced HCC [4,5,6,7]. Despite proposals from several groups, universally accepted guidelines regarding the selection of these patients are still lacking [8].

Pretransplant LRTs including TACE, radiofrequency ablation (RFA), and percutaneous ethanol injection (PEI) are regarded as adjunct tools to downstage advanced HCC. The very early stage (BCLC 0) is defined as the presence of a single nodule of <2 cm in diameter and without vascular invasion or metastases in patients with good performance status (PS 0) and well-preserved liver function (Child–Pugh A). The early stage (BCLC A) corresponds to patients with one nodule of <5 cm or up to three nodules each of <3 cm. Patients with BCLC stages 0 and A are candidates for potentially curative treatment options, i.e., surgical resection, LT, or local ablation. The treatment approach for BCLC-A patients varies according to tumor number and degree of liver function impairment. In the intermediate stage (BCLC-B), the magnitude of the tumor burden may be quite heterogeneous, and prognosis is also influenced by AFP concentration and the degree of liver function impairment even if still belonging to Child–Pugh class A. This individualized patient profile may also determine whether liver transplantation, transarterial chemoembolization (TACE), or systemic therapy should be used. [9]. It is increasingly used to control tumor growth when the waiting time is prolonged and serves as a tool to improve candidate selection [10]. Systemic therapy is a deferred option in HCC because of the strong and broad resistance of HCC to cytotoxic chemotherapy. The use of single agents in therapy is practically non-existent because of their low response [11]. Tumor downstaging, defined as a reduction in viable tumor burden by LRT to meet acceptable LT criteria, has been considered as a better alternative to simply expanding the tumor size limits for LT [12,13,14]. In principle, downstaging serves as a tool to select a subset of patients with favorable tumor biology who would respond to downstaging treatments and do well after LT [13]. LRT has also developed from the concept of bridging cases to the time of LT to the concept of neoadjuvant (necrotizing) treatment [15].

AFP is widely used to distinguish a subset of LT candidates with a reasonable life expectancy after LT [16,17], and it is also considered a good predictor of the response to LRT in many other strategies [18]. It has many defects but remains the only relevant biomarker routinely used in HCC patients’ management. HCC prognosis being correlated with AFP, this biomarker should play a role in the decision of the various therapeutic strategies for patients with curative or palliative HCC, and not only before liver transplantation. With mounting evidence on preoperative AFP levels of prognostic interest, it would be relevant to AFP modulate to the therapeutic management strategies of HCC and the BCLC guidelines to improve the treatment of HCC patients [19,20]. Many liver transplant centers globally incorporate AFP, with differences in cutoffs, into their listing criteria. Therefore, the optimal serum AFP level cutoff as an exclusion criterion for LT in pretransplant HCC patients has been discussed abundantly. In a previous study, the subset of patients outside the Milan criteria with low serum AFP levels (0–15 ng/mL) displayed improved post-LT survival [21]. The high end of the AFP level cutoff, ranging from 400 ng/mL to 1000 ng/mL [6,22,23], had a poor prognosis. Bhat et al. reported that AFP value after TACE was significantly associated with better overall survival after LT in limited cases [24]. Mounting evidence reveals that AFP > 1000 ng/mL manifested in HCC patients either within or outside MC portends reduced post-LT survival and considerable risk of HCC recurrence [22,25,26].

The reduction in AFP is also thought to indicate a good response to non-transplant treatment [27,28,29]. We hypothesized that the AFP response to LRT might predict tumor recurrence after LDLT. The aim of this study was to evaluate the predictive value of AFP response to LRT and use it to stratify the tumor recurrence risk of HCC patients after LDLT.

## 2. Materials and Methods

### 2.1. Patients

Between January 2000 and December 2016, 370 HCC patients with LRT who underwent LDLT at Kaohsiung Chang Gung Memorial Hospital were enrolled in this study. Patients with HCC and combined hepatocellular cholangiocarcinoma were included, but those with other liver malignancies were excluded. Although some developments have been made, the surgical techniques used on donors and recipients have mainly remained the same and are described in detail elsewhere [30,31]. Decision making for primary resection, LRT, and LDLT were conducted as previously described [2]. Treatment recommendations were according to Barcelona Clinic Liver Cancer (BCLC) Stage [9]. None of the patients in this study had received chemotherapy. Systemic therapy is a deferred option in HCC because of the strong and broad resistance of HCC to cytotoxic chemotherapy [11].

Acceptance for LDLT required the candidate to fit the UCSF criteria, in accordance with Taiwan National Health Insurance policy. In our experience, the contraindications of LRT as downstaging include metastasis, major vascular invasion, a history of HCC rupture, and infiltrative-type tumors. The level of AFP (over 1000 ng/mL) was not the contraindication. Necrotizing therapy is advised if possible before transplant, even if the tumor status is within the criteria, to minimize the possibility of tumor recurrence [15]. Therefore, patients would receive LRT if they did not have contraindications for LRT. Before transplantation, chest and abdominal CT, bone scan, and brain MRI were routinely performed to exclude metastasis.

### 2.2. Patient Grouping

The serum AFP level was measured through radioimmunoassay (normal value: <20 ng/mL). One hundred and seventy-seven (47.8%) patients whose serum AFP levels were within the normal reference range (<20 ng/mL) before and after LRT were assigned to the control group. One hundred and ninety-three (52.2%) patients with abnormal serum AFP levels (>20 ng/mL) before LRT were stratified into the following groups according to the AFP response to LRT: complete AFP response (CR), partial AFP response (PR), and no AFP response (NR). The definition of CR was that AFP levels returned to the normal reference range (<20 ng/mL) after LRT. In patients with persistent abnormal AFP levels after LRT, the AFP decline was calculated by subtracting the pretransplant AFP level from the pre-LRT-maximal AFP level and dividing the result by the pre-LRT (or maximal) AFP level. The cutoff value of AFP decline between the PR and NR groups for tumor recurrence was determined by c-statistics using a receiver operating characteristic (ROC) curve.

### 2.3. Immunosuppression and Follow-Up Protocol

Basiliximab (Simulect; Novartis Pharma AG, Switzerland) was intravenously administered (20 mg) twice. Steroid therapy consisted of intraoperative intravenous methylprednisolone (500 mg), followed by a 20 mg/day dosage, which was tapered down and withdrawn after 3 months if no acute cellular rejection occurred. Patients with stable vital signs and renal function were given tacrolimus (Prograf; Fujisawa, Killorglin, Ireland) at a dose to maintain trough levels at 5–10 ng/mL during the first week after LDLT. Mycophenolate mofetil (CellCept; Roche, Ponce, Puerto Rico) was continuously administered at 0.5–1 g/day. Patients diagnosed with unfavorable tumor histology (such as poorly differentiated HCC, combined hepatocellular cholangiocarcinoma, or presence of macrovascular or microvascular invasion) or who were initially beyond UCSF criteria were given sirolimus (Rapamune; Pfizer, New York, NY, USA) at a dose that was maintained at 4–12 ng/mL. No adjuvant therapy was given to prevent recurrence. During the follow-up, the dosage of immunosuppressants was intentionally minimized if liver function was normal or stable. Follow-up visits in the outpatient clinic were scheduled on a monthly basis for the first year after transplantation and then every 3 months thereafter. Tumor recurrence was assessed by monthly measurements of AFP and liver ultrasound every 3 months. Computed tomography or magnetic resonance imaging were arranged when they were clinically indicated.

### 2.4. Statistical Analysis

Continuous variables were expressed as the mean ± standard deviation (SD) or as the median (interquartile range, IQR) if the data were not normally distributed. Student’s *t*-test or the Mann–Whitney U test was used for comparisons between groups as appropriate. Analysis of variance or the Kruskal–Wallis test was used for comparisons among groups as appropriate. Categorical variables were expressed as percentages and were compared using the chi-square test or Fisher’s exact test. The cumulative recurrence rate was calculated, and survival analysis was performed with the Kaplan–Meier method, and survival was compared between groups using the log-rank test. A Cox proportional hazards regression model (forward stepwise selection) was used to assess predictors of tumor recurrence and overall survival. Univariable and multivariable hazard ratios (HRs) and their corresponding 95% confidence intervals (CIs) were calculated. Statistical significance was set at *p*-value < 0.05. ROC curve analysis was performed using a nonparametric method. All statistical analyses were performed using SPSS for Windows, version 22.0 (IBM, Armonk, NJ, USA).

### 2.5. Ethics Statement

The current study was approved by the Chang Gung Medical Foundation Institutional Review Board (approval number: 201701632A3). All methods were performed in accordance with the approved guidelines. Written informed consent was waived by the Chang Gung Medical Foundation Institutional Review Board due to the retrospective design of this study.

## 3. Results

### 3.1. Patient Characteristics and Their Association with Recurrence

The 370 enrolled HCC patients were 295 men and 75 women, with a median age of 55.4 years. The median follow-up after LDLT was 85.9 months (IQR, 55.5–115.0 months). HCC recurred in 47 (12.7%) patients at a median of 15.2 months after LDLT (IQR, 7.1–30.7 months), of which 26 (55.3%) were extrahepatic metastases, and 4 (8.5%) were early extrahepatic metastases which was defined as within 6 months after transplantation. The causes of mortality were HCC- (n = 35, 9.5%), non-HCC- (n = 32, 8.2%), and surgery-related (n = 7, 1.9%). The patients’ demographic characteristics and histopathological results of the explanted liver are shown in Table 1. Most patients were infected with hepatitis B and C viruses. Compared with the nonrecurrence group, the recurrence group had higher pretransplant AFP levels, more frequently underwent LRT, had higher tumor burdens, and had higher rates of microvascular invasion and viable tumors.

### 3.2. Cutoff Value of AFP Response to LRT

In patients with persistent abnormal AFP levels after LRT, the median AFP decline was 71.2% (IQR: 21.5–91.5%). To further stratify the risk of tumor recurrence with AFP decline, a ROC curve was used to determine the cutoff point between the PR and NR groups (Figure 1A). AFP decline demonstrated acceptable discrimination for predicting tumor recurrence, with an area under the ROC curve of 0.641 (*p* = 0.05). With an optimal cutoff value of 15%, the sensitivity and specificity of predicting tumor recurrence were 65.0% and 67.4%, respectively. The scatter plot in Figure 1B shows the distribution of pre-LRT (or maximal) and pretransplant AFP levels in the LRT groups.

### 3.3. Recurrence Rates According to Pre-LRT AFP and AFP-Response Groups

Figure 2 shows the recurrence rates of the four groups based on pre-LRT AFP and AFP response. Of the 177 (11.9%) patients with normal pre-LRT serum AFP levels (control group), 21 experienced recurrence after LDLT. Of the patients with abnormal pre-LRT serum AFP levels, 84 had AFP levels that returned to normal (CR group) after LRT. Their recurrence rate (4/84, 4.8%) was significantly lower (*p* = 0.001) than those with persistent abnormal AFP levels (22/109, 20.2%). In patients with an AFP decline of more than 15% (PR group), 8 of 67 (11.9%) patients experienced recurrence after LDLT. In patients with an AFP decline less than 15% (NR group), their recurrence rate was significantly higher (*p* = 0.006) than the PR group.

Table 2 shows the clinicopathologic characteristics of the four groups. The AFP-response groups had significant differences in microvascular invasion and American Joint Committee on Cancer (AJCC) T stage. The frequency of complete tumor necrosis was higher in the CR group; however, the AFP and pathologic response groups had only borderline significance. Regarding the correlation between the AFP-response groups and LRT, we found that the PR group had the highest percentage (41.8%) of pre-LRT-maximal AFP levels and the NR group had the lowest percentage (14.3%). The NR group also had the highest recurrence (*p* < 0.001) and mortality rates (*p* = 0.001) among the three groups. Consequently, the medium of recurrence months was significantly less than in the CR group (*p* = 0.004) (Table 2).

### 3.4. Cumulative Recurrence Rate and Overall Survival

The 5-year cumulative recurrence rates in the control, CR, PR, and NR groups were 11.7%, 4.9%, 10.7%, and 33.4%, respectively (Figure 3A). The CR group had excellent outcomes, while the NR group had the worst outcomes. The PR group obtained similar outcomes to the control group. The 5-year overall survival rates of the control, CR, PR, and NR groups were 81.5%, 94.0%, 88.0%, and 70.7%, respectively (Figure 3B). The CR group had excellent outcomes, while the NR group had the worst outcomes.

Univariable analysis demonstrated that NR was an unfavorable factor (HR 3.213, 95% CI 1.633–6.322, *p* = 0.001) for tumor recurrence (Table 3) and overall survival (HR 1.873, 95% CI 1.044–3.361, *p* = 0.035). CR was a favorable factor (HR 0.496, 95% CI 0.247–0.995, *p* = 0.049) for overall survival (Table 4). In multivariable analyses, the groups, according to LRT number, AFP response to LRT, and largest tumor size, independently predicted tumor recurrence; moreover, AFP response to LRT and AJCC T stage predicted overall survival (Table 3 and Table 4).

## 4. Discussion

This study suggested that AFP response to LRT can be used to stratify the tumor recurrence risk after LDLT in HCC patients with abnormal pre-LRT AFP levels. Patients with AFP levels that returned to normal after LRT had excellent outcomes after LDLT. Patients with an AFP decline of more than 15% after LRT achieved comparable outcomes to those in the non-LRT and normal AFP groups. In contrast, patients with an AFP decline of less than 15% had the worst outcomes after LDLT. The pretransplant selection criteria for HCC patients could be refined with the consideration of AFP response to LRT.

High AFP was usually associated with a poor prognosis after LT. High AFP before transplant was a critical condition. Merani et al. [32] demonstrated that patients with an AFP level of >400 ng/mL at the time of listing who were downstaged to AFP ≤ 400 ng/mL had significantly better intent-to-treat survival than patients failing to show a reduction in AFP to <400 ng/mL. They concluded that downstaging HCC patients with high AFP was feasible and leads to similar survival to that of patients with persistently low AFP levels.

Bhat et al. [24] evaluated 35 HCC patients beyond the Milan criteria who received carboplatin-based TACE before LT and found that the percentage decrease in AFP was a significant predictor for survival and that patients with an AFP decrease exceeding 50% had significantly better median survival rates. Mehta et al. [33] had a similar finding focusing on a high level of AFP (>1000 ng/mL). Our study stratified patients into three groups according to their post-LRT AFP levels. Patients with AFP levels returning to normal were classified as the CR group and had excellent outcomes. The percentage decrease in AFP was only used in patients with persistent abnormal AFP levels after LRT, and the cutoff value of a 15% AFP decline was determined with a ROC curve. Noticeably, there were 25 patients who had high levels of AFP (>1000 ng/mL) before LRT. Four of them belonged to the NR group and the rest were either in the PR or CR group. The mortality and recurrence rates of the NR group were 38.1% and 33.3%, respectively, and of the control group were 21.5% and 11.9%, respectively.

Pretransplant LRT for HCC has been shown to be a strategy to improve long-term survival [34]. The concept of necrotizing therapy has been applied in our center, in which LRT was used as a neoadjuvant therapy to achieve complete or major tumor necrosis [15]. In our cohort, none of the 69 patients with pathological complete tumor necrosis experienced recurrence; however, it could only be recognized in the post-transplant explanted liver. Our results demonstrated that patients with a complete AFP response to LRT (abnormal AFP levels returning to normal levels after LRT) had the lowest 5-year cumulative recurrent rate (4.9%) and excellent 5-year overall survival (94.0%). The CR group could be used as a predictor for excellent outcomes before transplantation.

Our study demonstrated that patients with no AFP response to LRT (NR group) had the highest 5-year cumulative recurrence rate (33.4%) and worst 5-year overall survival (70.7%). AFP was a comprehensive biomarker of tumor behavior. AFP promotes the growth, proliferation, and metastasis of HCC and prevents apoptosis and the escape of HCC from immune surveillance [35]. A high AFP level has been shown to be associated with poorer outcomes [36,37]. If the serum AFP level fails to decrease after LRT, it might be due to technical or anatomic issues, aggressive tumor biology, or undetected tumors present in other parts of the liver or outside the liver. In this study, our patients had a whole-body bone scan and brain MRI to exclude the possibility of extrahepatic metastasis. Only one patient in the NR group had extrahepatic metastasis. Our data showed that the NR group more frequently had microvascular invasion. Therefore, NR indicated that tumors had a more aggressive behavior, and NR could be used as a predictor for tumor recurrence and poor outcomes.

Our study also showed that patients in the PR group with a partial AFP response to LRT had significantly better outcomes than those in the NR group. Compared with patients with normal AFP levels (control group), there were no significant differences in the 5-year cumulative recurrent rate and overall survival. In patients with cirrhotic liver and impaired liver function, comprehensive LRT is intolerable. One cycle of LRT resulting in a 15% decrease in the AFP level might be able to predict promising outcomes after LT.

In some previous studies, suboptimal results with high recurrence rates were seen in patients who had received LT with an AFP level of ≥400 ng/mL [38,39]. We further investigated the details of three AFP-response groups with LRT. Notably, only 14.3% of maximal AFP in the NR group was higher than 400 ng/mL. The PR group had the highest proportion of maximal AFP among the three groups. This result indicated that a high level of AFP was not a main indicator of recurrence in this study. AFP response could be another indicator to predict the risk of recurrence. If AFP did not respond to the LRT, the prognosis was worse than for PR even if the AFP was still low. Nevertheless, if AFP could partially respond to LRT, even decreasing by only 15% to the maximal AFP, it could improve the outcome after LT. Recent studies suggested that the change in AFP while on the waiting list could best predict survival, with cutoff values of 200, 50, or 15 ng/mL/month in patients without downstaging or bridging LRT [40,41,42]. However, in our center, downstaging or bridging LRT was an important method to help patients fitting the national criteria. This study also showed that the AFP response could be used to distinguish the efficacy of LRT. The dynamic change in AFP was indicative of the tumor’s biological behavior, which was complicated. In patients with pretransplant LRT, a single AFP level could not be used for evaluating tumor aggressiveness. Therefore, AFP response to LRT was an alternative method for evaluating tumor aggressiveness and could be used to stratify the risk of tumor recurrence in HCC patients after LDLT. In this retrospective study, we simplified two points of AFP level to evaluate the prognosis. It was not only more accurate than a single point but also simpler than dynamic AFP levels. In a medical application, if LRT produces a complete or partial response before LDLT in abnormal AFP patients, a better prognosis can be expected. Although the survival rate in the no AFP response group was higher than that of most patients receiving hepatectomy or ablation, obviously the inferior survival compared with the AFP-response groups will change the post-transplant treatment policy. For instance, post-transplant management such as immunosuppressant modification and the indication of adjuvant therapy by target therapy (study on-going) to reduce recurrence are completely different. By the pretransplant AFP response from our study, we can stratify the patients for different post-transplant protocols [43].

In real-world practice, patients with high AFP (>1000 ng/mL) would be normally exclude by some centers. In our LRT groups, there were 11 patients (13.4%) in the CR group, 10 (14.9%) in the PR group, and 4 (9.5%) in the NR group whose maximal AFP was beyond 1000 ng/mL. The proportion in each group was similar, but the outcomes were very different.

The present study had some limitations. First, this study was a single-center retrospective study with a small sample size. Second, there was a lack of data on HCC-related dropouts from the waiting list and patient selection bias. In our center, infiltrative tumors or those accompanied by sky-high AFP as contraindication criteria for a downstaging procedure are excluded in the first place. Third, the pretransplant LRT method was not standardized. The choice of TACE, RFA, PEI, or their combined use depended on the patient’s clinical condition.

## 5. Conclusions

This study showed that AFP response to LRT was a predictor for tumor recurrence in HCC after LDLT. It was an easy and feasible method to stratify the risk of tumor recurrence. The CR group had better outcomes than the control group of patients with normal AFP levels. The NR group had the worst outcomes, whereas the outcomes of the PR group were comparable to those of the control group. These results might encourage patients who planned to undergo LT but could not reach a complete AFP response or had a high level of AFP in the first place. The patients who had high AFP (>400 ng/mL) might consider undergoing LRT and could still have the opportunity to get a better outcome. If they still do not have a partial AFP response after LRT, the surgeon can monitor the condition of these patients to treat early when recurrence occurs.

## Figures and Tables

**Figure 1 cancers-15-01551-f001:**
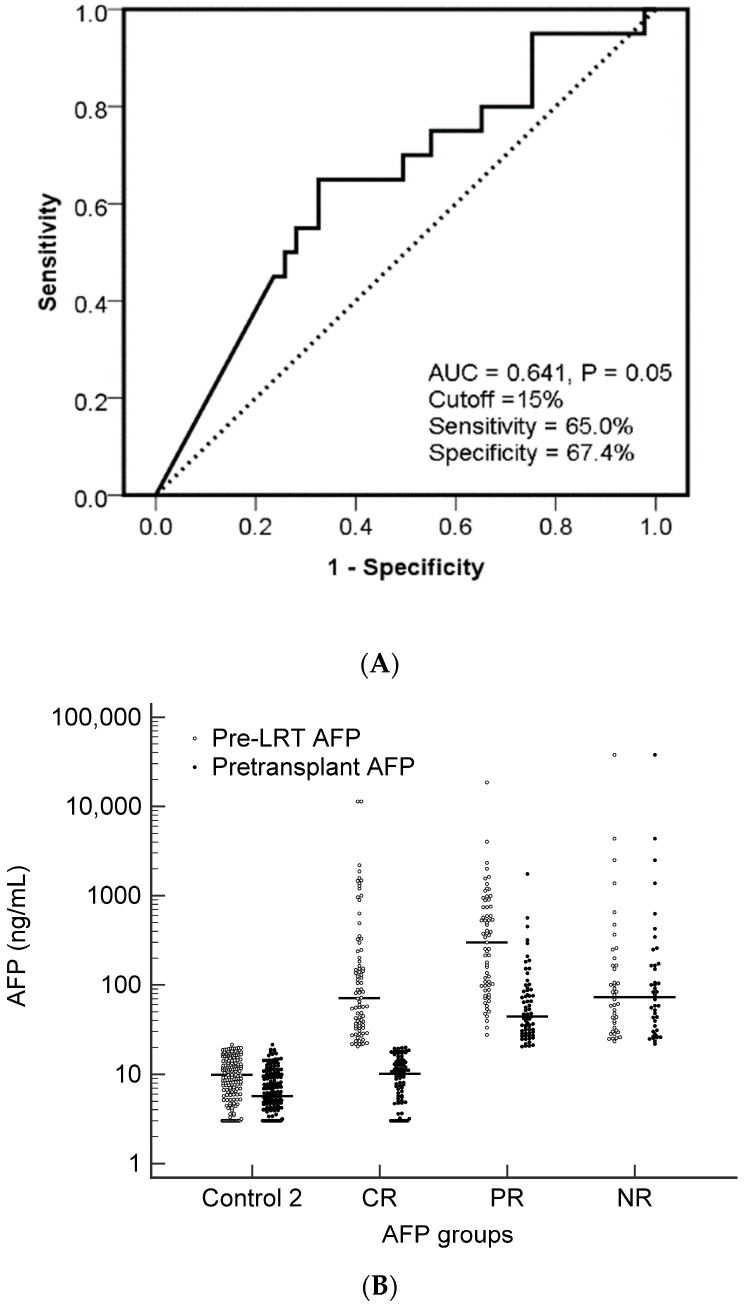
(**A**) Receiver operating characteristic curve analysis of AFP decline in patients with persistent abnormal AFP levels after locoregional therapy (LRT) for predicting tumor recurrence after living donor liver transplantation. (**B**). The distribution of the pre-LRT-maximal and pretransplant AFP levels in four defined groups: control, complete AFP response (CR), partial AFP response (PR), and no AFP response (NR).

**Figure 2 cancers-15-01551-f002:**
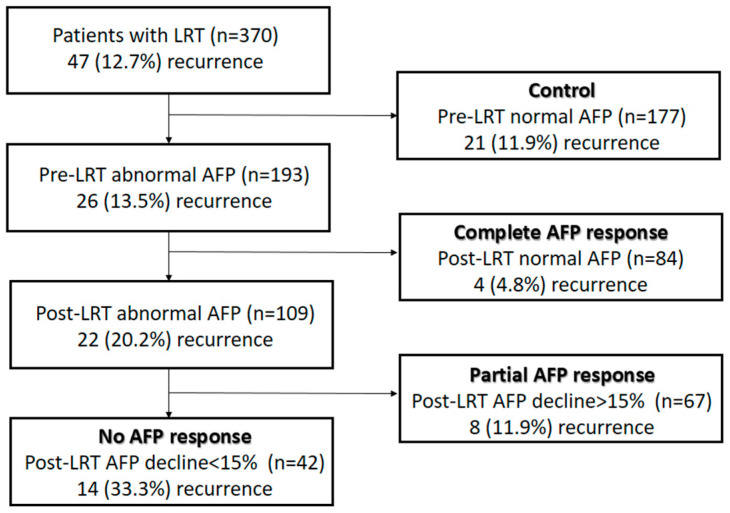
Flowchart of patient grouping according to AFP level, AFP response, and recurrence rates. The recurrence rates show the significant difference in the no AFP response group compared to the other three groups.

**Figure 3 cancers-15-01551-f003:**
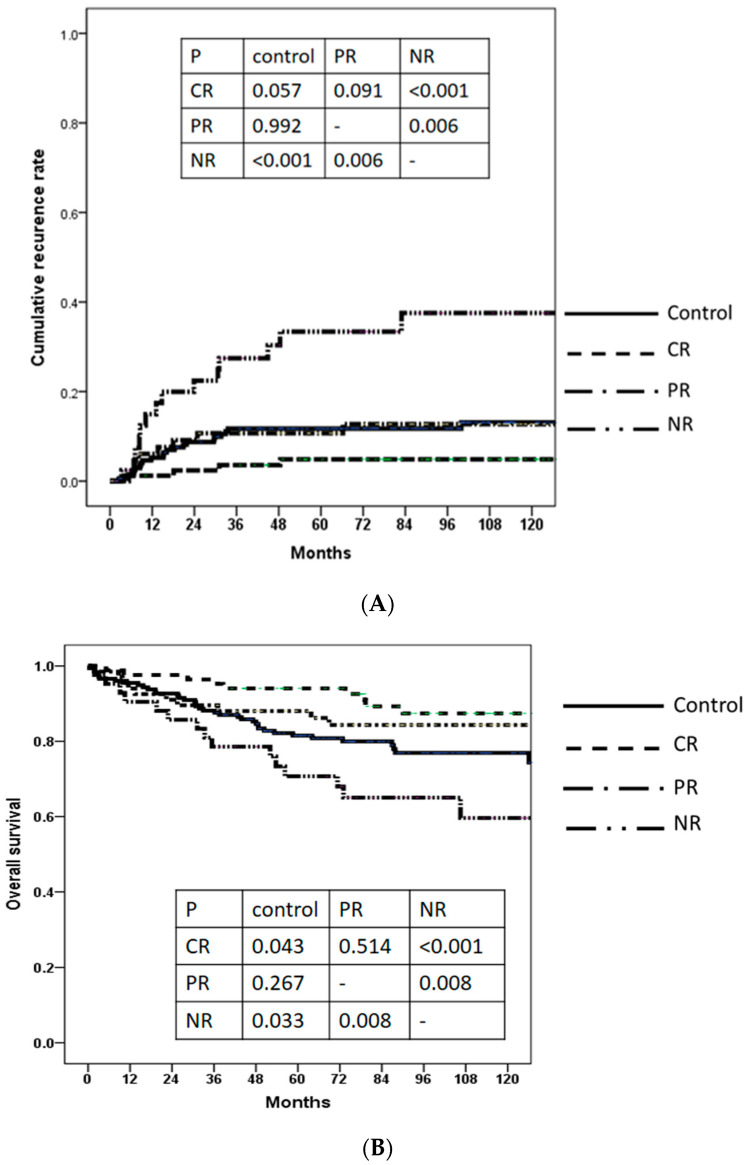
Cumulative recurrence rates (**A**) and overall survival (**B**) of the four groups according to AFP level and AFP response (CR, complete AFP response; PR, partial AFP response; NR, no AFP response). CR group showed the excellent cumulative recurrence rate and survival rate compared with NR group. PR group was comparable to the control.

**Table 1 cancers-15-01551-t001:** Clinicopathologic characteristics of patients associated with tumor recurrence after living donor liver transplantation.

Clinicopathologic Characteristics	All Patients (n = 370)	Nonrecurrence (n = 323)	Recurrence (n = 47)	*p*
Age (years), median (IQR)	55.4 (50.9–60.7)	55.5 (50.2–60.7)	54.7 (50.5–60.1)	0.609
Gender, male, n (%)	295 (79.7)	254 (78.6)	41 (87.2)	0.171
BMI, median (IQR)	25.2 (22.8–27.5)	25.2 (22.8–27.5)	25.3 (23.1–27.4)	0.846
Liver disease, n (%)				
HBV	226 (61.1)	193 (59.8)	33 (70.2)	0.169
HCV	135 (36.5)	122 (37.8)	13 (27.7)	0.178
Alcoholic	24 (6.5)	20 (6.2)	4 (8.5)	0.753
Others	6 (1.6)	6 (1.9)	-	0.346
Child–Pugh class, n (%)				0.963
A	204 (55.1)	179 (55.4)	25 (53.2)	
B	23(33.2)	105 (32.5	18 (38.3)	
C	43(11.6)	39 (12.1)	4 (8.5)	
MELD, median (IQR)	9.0 (7.0–12.0)	9.0 (7.0–12.0)	9.0 (7.0–12.0)	0.996
Pre-LRT-maximal AFP (ng/mL), median (IQR)	23.5 (10.8–109.9)	22.0 (10.7–106.0)	30.0 (11.3–200.0)	0.396
Pre-LRT-maximal AFP > 20 ng/mL, n (%)	193 (52.2)	167 (51.7)	26 (55.3)	0.643
Pretransplant AFP (ng/mL), median (IQR)	10.5 (5.5–26.5)	10.2 (5.5–22.9)	12.7 (5.8–75.1)	0.050
Pretransplant AFP > 20 ng/mL, n (%)	110 (29.7)	87 (27.0)	23 (47.6)	0.002
Pretransplant LRT number, median (IQR)	2 (1–3)	2 (1–3)	3 (2–5)	<0.001
Pretransplant LRT methods, n (%)				0.003
RFA only	74 (20.0)	72 (22.3)	2 (4.3)	
TACE only	128(34.6)	114 (35.3)	14 (29.8)	
PEI only	9 (2.4)	7 (2.2)	2 (4.3)	
TACE and RFA	97 (26.2)	77 (23.8)	20 (42.6)	
TACE and PEI	28 (7.6)	24 (7.4)	4 (8.5)	
RFA and PEI	8 (2.2)	7 (2.2)	1 (2.1)	
TACE, RFA, and PEI	26 (7.0)	22 (6.8)	4 (8.5)	
Tumor number, median (IQR)	2 (1–3)	2 (1–3)	2 (2–4)	0.067
Largest tumor size (cm), median (IQR)	2.7 (2.0–3.5)	2.6 (2.0–3.5)	3.0 (2.5–5.0)	0.001
Total tumor size (cm), median (IQR)	4.5 (2.8–7.0)	4.3 (2.6–6.5)	6.5 (3.8–11.0)	<0.001
Beyond Milan criteria, n (%)	168 (45.4)	138 (42.7)	30 (63.8)	0.007
Beyond UCSF criteria, n (%)	122 (33.0)	95 (29.4)	27 (57.4)	<0.001
Microvascular invasion, n (%)	104 (28.1)	73 (22.6)	31 (66.0)	<0.001
AJCC T stage, n (%)				<0.001
T1	96 (25.9)	94 (29.1)	2 (4.3)	
T2	250 (67.6)	213 (65.9)	37 (78.7)	
T3	18 (4.9)	13 (4.0)	5 (10.6)	
T4	6 (1.6)	3 (0.9)	3 (6.4)	
Tumor necrosis, n (%)				0.050
No tumor necrosis	31 (8.4)	30 (9.3)	1 (2.1)	
Partial tumor necrosis	270 (73.0)	224 (69.3)	46 (97.9)	
Complete tumor necrosis	69 (18.6)	69 (21.4)	0 (0)	

**Table 2 cancers-15-01551-t002:** Clinicopathologic characteristics of patients in the 4 groups according to locoregional therapy, pre-LRT AFP, and AFP response.

Clinicopathologic Characteristics	Control (n = 177)	Complete AFP Response (n = 84)	Partial AFP Response (n = 67)	No AFP Response (n = 42)	*p* *	*p* **
Age (years), median (IQR)	55.5 (51.3–60.6)	55.5 (49.5–60.7)	54.3 (49.0–59.9)	55.5 (51.6–60.8)	0.703	0.599
Male (%)	146 (82.5)	64 (76.2)	54 (80.6)	31 (73.8)	0.256	0.683
BMI, median (IQR)	25.5 (22.7–27.7)	24.8 (23.1–27.3)	25.2 (22.5–27.5)	25.1 (22.6–27.3)	0.809	0.986
Liver disease (%)						
HBV	111 (62.7)	56 (66.7)	34 (50.7)	25 (59.5)	0.226	0.141
HCV	55 (31.1)	31 (36.9)	29 (43.3)	20 (47.6)	0.120	0.480
Alcoholic	13 (7.3)	5 (6.0)	4 (6.0)	2 (4.8)	0.921	0.957
Others	5 (2.8)	0	1 (1.5)	0	0.183	0.389
Child–Pugh class, n (%)					0.576	0.370
A	100 (56.5)	45 (53.6)	40 (59.7)	19 (45.2)		
B	54 (30.5)	33 (39.3)	19 (28.4)	17 (40.5)		
C	23 (13.0)	6 (7.1)	8 (11.9)	6 (14.3)		
MELD, median (IQR)	9 (7–12)	9.5 (7–12)	10 (7–12)	11 (8–13.3)	0.132	0.206
Maximal AFP (ng/mL), median (IQR)	9.9 (6.0–14.6)	71.2 (33.4–229.7)	301.0 (97.3–743.0)	72.7 (32.6–174.5)	<0.001	<0.001
Maximal AFP >400 ng/mL, n (%)	-	15 (17.2%)	28 (41.8%)	6 (14.3%)	<0.001	0.001
Pretransplant AFP (ng/mL), median (IQR)	5.7 (3.5–9.7)	10.2 (5.2–14.3)	44.2 (28.9–87.0)	72.7 (29.1–88.1)	<0.001	<0.001
Pretransplant AFP > 400, n (%)	-	-	3 (4.5)	6 (14.3)	<0.001	0.002
pretransplant LRT number, median (IQR)	2 (1–4)	2 (1–3)	2 (1–3)	3 (1–4)	0.327	0.239
Pretransplant LRT methods, n (%)					0.102	0.195
TACE only	56 (31.6)	25 (29.8)	32 (47.8)	15 (35.7)		
RFA only	36 (20.3)	26 (31.0)	8 (11.9)	4 (9.5)		
PEI only	5 (2.8)	3 (3.6)	0	1 (2.4)		
TACE and RFA	44 (24.9)	19 (22.6)	21 (31.3)	13 (31.0)		
TACE and PEI	17 (9.6)	6 (7.1)	3 (4.5)	2 (4.8)		
RFA and PEI	4 (2.3)	1 (1.2)	1 (1.5)	2 (4.8)		
TACE, RFA, and PEI	15 (8.5)	4 (4.8)	2 (3.0)	5 (11.9)		
Tumor number, median (IQR)	2 (1–3)	2.0 (1–3)	2 (1–4)	2.5 (2–5)	0.204	0.111
Largest tumor size (cm), median (IQR)	2.6 (2.0–3.5)	3.0 (2.1–3.8)	2.7 (2.0–3.4)	2.8 (2.0–3.6)	0.678	0.769
Total tumor size (cm), median (IQR)	4.3 (2.7–6.3)	4.5 (2.9–6.9)	4.2 (2.7–8.0)	6.1 (3.2–10.0)	0.179	0.168
Beyond Milan criteria, n (%)	73 (41.2)	38 (45.2)	32 (47.8)	25 (59.5)	0.039	0.305
Beyond UCSF criteria, n (%)	56 (31.6)	25 (29.8)	23 (34.3)	18 (42.9)	0.485	0.344
Microvascular invasion, n (%)	42(23.7)	18(21.4)	22(32.8)	22(52.4)	0.001	0.002
AJCC T stage, n (%)					0.040	0.029
T1	53 (29.9)	24 (28.6)	16 (23.9)	3 (7.1)		
T2	112 (63.3)	57 (67.9)	45 (67.2)	36 (85.7)		
T3	8 (4.5)	3 (3.6)	5 (7.5)	2 (4.8)		
T4	4 (2.3)	0	1 (1.5)	1 (2.4)		
Tumor necrosis, n (%)					0.07	0.087
No tumor necrosis	19 (10.7)	5 (6.0)	7 (10.4)	4 (9.5)		
Partial tumor necrosis	119 (67.2)	59 (70.2)	55 (82.1)	33 (78.6)		
Complete tumor necrosis	39 (22.0)	20 (23.8)	5 (7.5)	5 (11.9)		
Recurrence, n (%)	21 (11.9)	4 (4.8)	8 (11.9)	14 (33.3)	<0.001	<0.001
Recurrence, n (%)	21 (11.9)	4 (4.8)	8 (11.9)	14 (33.3)	<0.001	<0.001
Recurrence model					0.776	0.612
Intra-hepatic metastasis	10 (47.6)	1 (25.0)	3(37.5)	7(50.0)		
Extrahepatic metastasis	11 (52.4)	3 (75.0)	5 (62.5)	7 (50.0)		
Early extrahepatic metastasis (<6 months), n (%)	2 (1.1)	0 (0.0)	1 (1.5)	1 (2.4)	-	-
Recurrence months, median (IQR)	15.2 (7.8–29.60)	24.6 (17.6–44.1)	10.9 (6.4–23.1)	13.9 (8.0–34.5)	0.736	0.594
RFS months, median (IQR)	76.1 (49.1–112.3)	95.2 (73.0–121.5)	89.7 (56.5–118.2)	69.8 (21.6–104.2)	0.003	0.004
Expired, n (%)	38 (21.5)	10 (11.9)	10 (14.9)	16 (38.1)	0.004	0.001
OS months, median (IQR)	76.1 (50.7–112.7)	95.2 (74.0–121.5)	90.2 (59.5–118.2)	75.6 (41.0–110.0)	0.033	0.085

Control: patients with normal AFP levels before and after LRT. Complete AFP response: patients with abnormal AFP levels before LRT and normal AFP levels after LRT. Partial AFP response: patients with persistent abnormal AFP levels after LRT and the AFP level declines by more than 15%. No AFP response: patients with persistent abnormal AFP levels after LRT and the AFP level declines by less than 15%. * Comparison among 4 groups. ** Comparison among 3 groups of complete AFP response, partial AFP response, and no AFP response.

**Table 3 cancers-15-01551-t003:** Cox hazard analysis (univariable and multivariable) of clinicopathologic data for tumor recurrence with locoregional therapy.

Clinicopathologic Characteristics	HR (95% CI) Univariable	*p*	HR (95% CI) Multivariable	*p*
Age (years)	0.996 (0.962–1.032)	0.842		
Gender, male	1.724 (0.732–4.061)	0.213		
BMI	0.968 (0.892–1.051)	0.439		
Liver disease				
HBV	1.478 (0.791–2.763)	0.221		
HCV	0.672 (0.355–1.273)	0.223		
Alcoholic	1.352 (0.485–3.769)	0.564		
Others	0.049 (0.00–1614.0)	0.569		
Child–Pugh class		0.671		
A	1			
B	1.196 (0.652–2.192)	0.563		
C	0.755 (0.263–2.169)	0.601		
MELD	1.021 (0.971–1.072)	0.421		
Log10 maximal AFP (ng/mL)	1.135 (0.800–1.610)	0.478		
Log10 pretransplant AFP (ng/mL)	1.743 (1.154–2.630)	0.008		
Pretransplant LRT number	1.108 (1.042–1.178)	0.001	1.098 (1.017–1.186)	0.016
Pretransplant LRT methods		0.091		
RFA only				
TACE only	4.299 (0.977–18.92)	0.054		
PEI only	7.974 (1.123–56.62)	0.038		
TACE and RFA	8.660 (2.024–37.06)	0.004		
TACE and PEI	5.557 (1.018–30.342)	0.048		
RFA and PEI	4.582 (0.415–50.544)	0.214		
TACE, RFA, and PEI	6.025 (1.103–32.900)	0.038		
Group (AFP response)		<0.001		0.004
Normal AFP	1		1	
Complete AFP response	0.369 (0.126–1.074)	0.067	0.289 (0.096–0.870)	0.027
Partial AFP response	1.000 (0.443–2.258)	1.000	0.951 (0.418–2.164)	0.905
No AFP response	3.213 (1.633–6.322)	0.001	2.272 (1.115–4.631)	0.024
Pathological tumor characteristics				
Tumor number	1.142 (1.042–1.252)	0.005		
Largest tumor size (cm)	1.596 (1.322–1.927)	<0.001	1.515 (1.194–1.923)	0.001
Total tumor size (cm)	1.137 (1.079–1.198)	<0.001		
Beyond Milan criteria	2.176 (1.200–3.946)	0.010		
Beyond UCSF criteria	2.936 (1.646–5.236)	0.001		
AJCC T stage		<0.001		
T1	1			
T2	7.462 (1.798–30.97)	0.006		
T3	17.77 (3.445–91.61)	0.001		
T4	38.28 (6.387–229.4)	<0.001		

**Table 4 cancers-15-01551-t004:** Cox hazard analysis (univariable and multivariable) of clinicopathologic data for overall survival.

Clinicopathologic Characteristics	HR (95% CI) for Univariable	*p*	HR (95% CI) for Multivariable	*p*
Age (years)	1.000 (0.972–1.029)	0.975		
Gender, male	0.923 (0.523–1.628)	0.782		
BMI	0.976 (0.914–1.041)	0.454		
Liver disease				
HBV	0.895 (0.561–1.431)	0.644		
HCV	1.130 (0.706–1.809)	0.611		
Alcoholic	0.929 (0.339–2.548)	0.886		
Others	0.906 (0.126–6.525)	0.922		
Child–Pugh class		0.574		
A	1			
B	1.234 (0.753–2.024)	0.404		
C	1.364 (0.677–2.749)	0.386		
MELD	1.013 (0.974–1.055)	0.517		
Log10 maximal AFP (ng/mL)	1.012 (0.756–1.355)	0.936		
Log10 pretransplant AFP (ng/mL)	1.491 (1.053–2.112)	0.024		
Pretransplant LRT number	1.065 (1.004–1.131)	0.038		
Pretransplant LRT methods		0.220		
RFA only	1			
TACE only	1.737 (0.781–3.861)	0.176		
PEI only	0.861 (0.108–6.897)	0.888		
TACE and RFA	2.379 (1.068–5.296)	0.034		
TACE and PEI	3.166 (1.244–8.059)	0.016		
RFA and PEI	1.064 (0.133–8.510)	0.953		
TACE, RFA, and PEI	1.795 (0.587–5.488)	0.305		
Group (AFP response)		0.005		0.010
Normal AFP	1			
Complete AFP response	0.496 (0.247–0.995)	0.049	0.523 (0.260–1.052)	0.069
Partial AFP response	0.665 (0.331–1.334)	0.251	0.667 (0.331–1.341)	0.255
No AFP response	1.873 (1.044–3.361)	0.035	1.834 (1.000–3.365)	0.050
s				
Tumor number	1.034 (0.939–1.139)	0.497		
Largest tumor size (cm)	1.193 (1.010–1.410)	0.038		
Total tumor size (cm)	1.047 (0.992–1.105)	0.096		
Beyond Milan criteria	1.072 (0.678–1.695)	0.766		
Beyond UCSF criteria	1.353 (0.845–2.169)	0.208		
AJCC T stage		<0.001		0.001
T1	1		1	
T2	1.390 (0.766–2.522)	0.278	1.258 (0.685–0.307)	0.459
T3	3.745 (1.564–8.964)	0.003	9.403 (2.816–31.40)	0.001
T4	6.984 (2.291–21.29)	0.001	-	0.982

## Data Availability

Data are contained within the article.

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
