# Peer review of "AFP Response to Locoregional Therapy Can Stratify the Risk of Tumor Recurrence in HCC Patients after Living Donor Liver Transplantation"

_cancers, 2023, doi:10.3390/cancers15051551_

Round 1
Reviewer 1 Report
This study aimed to investigate the value of AFP response to pre-transplantion treatment for HCC patients. There is lack of predict markers to critical information for treatment strategy before transplantation. However, the following weakness should be noted:
1. Whether PET/CT was performed for all these patients? Extra-hepatic metastases limited the AFP response to local regional therapy. Also, several parameters should be added, including the recurrence model, especially for patients suffered early extra-hepatic metastases.
2. For those with AFP non response to LRT had 5-year overall survival of 70.7%, which was higher than most of patients receiving hepatectomy or ablation. Therefore, identifying those with worst survival outcome, is still could not change the treatments.
3. “AFP is a comprehensive biomarker of tumor behavior. AFP promotes the growth, proliferation, and metastasis of HCC and prevents apoptosis and the escape of HCC from immune surveillance.” Please provide references to support it.
4 As the authors indicated that AFP response preformed than the image response in the discussion section, please provide more detailed data in the results section.
5 There are several careless grammatical, tense, or punctuation errors in the manuscript, like “We also compared to image response with AFP response and found AFP response was more accurate that image response”.
Reviewer 2 Report
Abstract needs to be concise and clear with focused conclusion on finding in this case AFP
In the Introduction and Discussion Elaborate more on AFP and Hepatocellular carcinoma , therapy clinics and marker during and after therapy, Elaborate more on Chemotherapy on the Hepatocellular carcinoma
Elaborate more on pretransplant locoregional therapies in the Introduction and Discussion
Figure 1 2 and 4 to be one figure
All Figure Legend need to bi in detailed explanation
